# The Sleep Quality- and Myopia-Linked PDE11A-Y727C Variant Impacts Neural Physiology by Reducing Catalytic Activity and Altering Subcellular Compartmentalization of the Enzyme

**DOI:** 10.3390/cells12242839

**Published:** 2023-12-14

**Authors:** Irina Sbornova, Emilie van der Sande, Snezana Milosavljevic, Elvis Amurrio, Steven D. Burbano, Prosun K. Das, Helen H. Do, Janet L. Fisher, Porschderek Kargbo, Janvi Patel, Latarsha Porcher, Chris I. De Zeeuw, Magda A. Meester-Smoor, Beerend H. J. Winkelman, Caroline C. W. Klaver, Ana Pocivavsek, Michy P. Kelly

**Affiliations:** 1Department of Anatomy & Neurobiology, University of Maryland School of Medicine, 20 Penn St., Baltimore, MD 21201, USApdas1@umbc.edu (P.K.D.); janvipatel@som.umaryland.edu (J.P.);; 2Department of Ophthalmology, Erasmus Medical Center, Wytemaweg 40, 3015 CN Rotterdam, The Netherlands; 3Department of Epidemiology, Erasmus Medical Center, Wytemaweg 40, 3015 CN Rotterdam, The Netherlands; 4The Netherlands Institute for Neuroscience (NIN), Royal Dutch Academy of Art & Science (KNAW), Meibergdreef 47, 1105 AZ Amsterdam, The Netherlands; 5Department of Pharmacology, Physiology & Neuroscience, University of South Carolina School of Medicine, Garners Ferry Rd., Columbia, SC 29209, USA; 6Department of Neuroscience, Erasmus Medical Center, Wytemaweg 40, 3015 CN Rotterdam, The Netherlands; 7Department of Ophthalmology, Radboud University Medical Center, Geert Grooteplein Zuid 10, 6525 GA Nijmegen, The Netherlands; 8Institute of Molecular and Clinical Ophthalmology, Mittlere Strasse 91, 4070 Basel, Switzerland; 9Center for Research on Aging, University of Maryland School of Medicine, 20 Penn St., Baltimore, MD 21201, USA

**Keywords:** phosphodiesterase, sleep, myopia, eye, cAMP, cGMP, trafficking, Golgi, PDE11A

## Abstract

Recently, a Y727C variant in the dual-specific 3′,5′-cyclic nucleotide phosphodiesterase 11A (PDE11A-Y727C) was linked to increased sleep quality and reduced myopia risk in humans. Given the well-established role that the PDE11 substrates cAMP and cGMP play in eye physiology and sleep, we determined if (1) PDE11A protein is expressed in the retina or other eye segments in mice, (2) PDE11A-Y7272C affects catalytic activity and/or subcellular compartmentalization more so than the nearby suicide-associated PDE11A-M878V variant, and (3) *Pde11a* deletion alters eye growth or sleep quality in male and female mice. Western blots show distinct protein expression of PDE11A4, but not PDE11A1-3, in eyes of *Pde11a* WT, but not KO mice, that vary by eye segment and age. In HT22 and COS-1 cells, PDE11A4-Y727C reduces PDE11A4 catalytic activity far more than PDE11A4-M878V, with both variants reducing PDE11A4-cAMP more so than PDE11A4-cGMP activity. Despite this, *Pde11a* deletion does not alter age-related changes in retinal or lens thickness or axial length, nor vitreous or anterior chamber depth. Further, *Pde11a* deletion only minimally changes refractive error and sleep quality. That said, both variants also dramatically alter the subcellular compartmentalization of human and mouse PDE11A4, an effect occurring independently of dephosphorylating PDE11A4-S117/S124 or phosphorylating PDE11A4-S162. Rather, re-compartmentalization of PDE11A4-Y727C is due to the loss of the tyrosine changing how PDE11A4 is packaged/repackaged via the trans-Golgi network. Therefore, the protective impact of the Y727C variant may reflect a gain-of-function (e.g., PDE11A4 displacing another PDE) that warrants further investigation in the context of reversing/preventing sleep disturbances or myopia.

## 1. Introduction

3′,5′-cyclic nucleotide phosphodiesterases (PDEs) are often-investigated drug targets due to their ability to modulate cyclic nucleotides in an organ and brain region-specific manner, as well as a subcellular compartment-specific manner [1]. There are 11 families of PDEs (PDE1-11), each composed of multiple isoforms that show unique tissue expression profiles and distinctive subcellular distributions [1]. Thus, each PDE isoform is a unique drug target capable of modulating separable physiological processes. The PDE11A family breaks down both cAMP and cGMP [2] and is comprised of four spliciforms: PDE11A1, A2, A3, and A4. The longest isoform, PDE11A4, is the isoform that is expressed in neural tissue [2], and it is ~95% homologous across mouse, rat, and human [2]. This high degree of homology argues that the impact of human genetic variants can be readily interrogated in rodent systems. PDE11A4 is strongly expressed in neurons of the ventral and dorsal hippocampus [3,4], the adjacent amygdalohippocampal and amygdalostriatal transition areas [5], the spinal cord, and the dorsal root ganglion [2], but not in any of the 20+ peripheral organs we have compared in young adult *Pde11a* KO versus WT mice to date [6]. Consistent with its particularly enriched expression in the ventral hippocampus, the best-known neurological roles for PDE11A4 are regulating social interactions, altering the formation of social memories, impacting neuroinflammation, and triggering age-related cognitive decline [3,4,5,7,8,9,10,11,12].

Interestingly, a Y727C missense mutation in PDE11A has been genetically linked in humans to improved sleep quality and a reduced risk of myopia [13,14,15,16]. Ventral hippocampus activity in animals and humans correlates with sleep processes [17,18], and aberrant sleep is associated with hippocampal dysfunction at a molecular, cellular, and behavioral level [19,20,21,22]. Further, ventral hippocampal projections containing PDE11A4 project to the hypothalamus [5], a brain region well known to regulate sleep [23]. Although the presence of PDE11A protein has not yet been reported in the eye, its presence has been implied by human single-cell RNA sequencing studies [24,25,26,27,28]. Further, myopic development is associated with inadequate exposure to light [13,29,30], photo-transduction is highly dependent on hydrolysis of the PDE11A substrate cGMP [31], and specific wave lengths of light that upregulate PDE11A expression in fibroblast cells [32] demonstrate promise as potential anti-myopic agents [29]. Of particular interest to the fact that PDE11A-Y727C is both protective against myopia and enhancing to sleep quality, myopia has been associated with reduced sleep quality [33,34] and a dysregulated retinal circadian clock [35,36]. In addition, RNA sequencing and phosphoproteomic studies show PDE11A4 regulates the molecular underpinnings of circadian entrainment in the ventral hippocampus [5]. Together, these studies suggest the PDE11A-Y727C variant stands to impact sleep quality and eye physiology.

Given this evidence, along with the well-established role for cyclic nucleotide signaling and PDE function in eye physiology [36,37,38,39,40,41,42,43,44,45,46,47] and sleep [19], the current study determined (1) if PDE11A is expressed in the mouse retina, the anterior segment, and/or posterior segment of the mouse eye and if that expression changes with age, (2) if the PDE11A-Y727C mutation affects PDE11A catalytic activity or trafficking more so than the nearby PDE11A-M878V variant that is associated with high-risk suicide [48] but not associated with myopia or sleep, and (3) if the loss of PDE11A alters ocular biometry, refractive error, or sleep quality.

## 2. Methods

### 2.1. Subjects

Studies here employed 3 lines of *Pde11a* knockout (KO) mice. The first line (used in analyses of segment-specific PDE11A4 expression and ocular measures) is the C57BL/6NCrl-PDE11A^em1(IMPC)Mbp/Mmucd^ from the Mutant Mouse Resource and Research Center (donating investigator: Kent Lloyd, University of California). In this mouse strain on a C57BL/6N background, exon 6 and flanking splicing regions were constitutively deleted from the *Pde11a* gene using Crispr Cas9 gene-editing technology (Crispr B6/N). The second and third lines (used in analyses of whole-eye PDE11A4 expression, cataract studies, and sleep studies) are the LacZ-generated Deltagen *Pde11a* KO lines that are maintained on either a predominantly C57BL/6J (LacZ B6/J) or BALB/cJ background (LacZ BALB/cJ), as previously published [12]. For each line, mice heterozygous for the *Pde11a* allele were mated to generate *Pde11a* homozygous KO and wild-type (WT) mice. The mice were socially housed and had access to water and food ad libitum. The studies were either approved by the University of Maryland, Baltimore and the University of South Carolina Institutional Care And Use Committees with the treatment and care conducted as per the National Institutes of Health Guide for the Care and Use of Laboratory Animals (Pub 85-23, revised 1996) or the ethical committee of the Royal Netherlands Academy of Arts and Sciences (KNAW, Amsterdam, The Netherlands) with the treatment and care of the animals performed in compliance with the ARVO Statement for the Use of Animals in Ophthalmic and Vision Research, as well as the European Communities Council Directive 2010/63/EU.

### 2.2. Tissue Isolation for Western Blot

All 3 lines of *Pde11a* KO mice were used in the analyses of PDE11A4 expression across the eye. The mice were killed by cervical dislocation with or without isoflurane anesthesia. The eyes were harvested fresh and kept whole or dissected in phosphate-buffered saline on wet ice. The ocular tissue was separately isolated for the (1) anterior segment (i.e., cornea, ciliary body and muscle, and zonules), (2) retina, and (3) posterior segment (i.e., sclera, choroid, Bruch’s membrane, retinal pigmented epithelium, and parts of the optic nerve). The tissue was pooled for both eyes of the same mouse, put on dry ice, and kept at −80 degrees until further processing. All the tissue samples were isolated between 8 a.m. and 2 p.m.

### 2.3. Preparation of Samples for Western Blots

All tissue samples were stored at −80 °C and kept on dry ice until processed. For analysis of whole-eye tissue, 215 μL of either ice-cold 20 mM Tris-HCl/2 mM MgCl_2_/0.5% Triton X-100 with protease inhibitor (Pierce #A32953) and phosphatase inhibitor cocktail 3 (#P0044 Sigma ST Louis, MO, USA; Buffer #1, Bf#1) or boiling 1%SDS/50 mM NaF (Buffer #2, Bf#2) was added. Due to the tough nature of the eyes, the samples then underwent three cycles of sonication (FB120 sonic dismembrator, Fisher, Waltham, MA, USA), where each cycle was followed by either freezing (Buffer #1) or boiling (Buffer #2) for five minutes. The samples were then pulse centrifuged, and the liquid portion was transferred to a new test tube.

For analysis of individual eye segments, all the samples were sonicated for 3 cycles using ice-cold Buffer #1. The homogenized retina samples were precleared by centrifuging at 4 °C for 10 min at 1000× *g* and transferring the resultant supernatant to a new test tube. Any solid tissue that remained in the anterior and posterior segment samples was shaken down to the bottom of the tube, and the liquid portion was transferred to a new test tube because preclearing greatly reduced the protein concentrations of the samples. Previously verified hippocampal tissue from either LacZ C57BL/6J or Crispr C57BL/6N *Pde11a* KO and WT mice were sonicated in Buffer #1 and used as negative and positive controls, respectively. To enable loading of the same positive and negative controls across many blots, hippocampal tissue from multiple male and female WT or KO mice were combined into one test tube.

The total protein concentration for each sample was determined using the DC Protein Assay kit (BioRad, Inc.; Hercules, CA, USA) according to the manufacturer’s protocol. The whole-eye samples and hippocampal controls were prepared at a concentration of 3.0 μg/μL. Due to the large number of samples being processed, the eye segment tissue was prepared in separate batches of WT and KO mice. A “batch” represented a set of samples that included a balanced number of subjects/biological group that could all fit on one gel. To increase the opportunity of detecting a PDE11A4 signal, each batch was prepared at the highest possible concentration (Table 1). Only 1 batch was loaded per gel, so any difference in the overall expression between gels due to differences in protein concentration were normalized by expressing data as a fold change of a given gel’s control group (see more below in Section 2.12). Postnatal day 7 (P7) samples could only be analyzed for the retina, as the other segments were insufficiently concentrated. Due to initially low concentrations, 2 batches underwent an additional step of acetone precipitation (as per Thermo Fisher Scientific protocol TR0049.1; see Table 1), which a pilot study confirmed did not interfere with PDE11A4 detection (Appendix A). The samples for Western blotting were prepared from the tissue lysates with the addition of Invitrogen sample buffer #NP0007 and sample-reducing agent #NP0009. The samples were then stored at −80 °C until processing by Western blot.

### 2.4. Western Blotting

Westerns blots were performed as previously described in [5,12,49]. To analyzePDE11A expression, 11 μL of each sample (see Table 1 for concentrations) was loaded onto 4–12% Bis-Tris NuPAGE polyacrylamide gradient gels (Life Technologies; Bedford, MA, USA) for electrophoresis. The protein was transferred onto a 0.45 μm nitrocellulose blotting membrane (#10600008; Amersham, Buckinghamshire, UK), which was then stained with Ponceau S (#6266-79-5Fisher Scientific, Waltham, MA USA). Ponceau S was used as a loading control based on the best practice statement of the *Journal of Biological Chemistry* [50]. The membranes were blocked in 5% milk/0.1% Tween20. The whole-eye blots were probed overnight at 4 °C with one of two PDE11A antibodies: the pan-PDE11A antibody PD11-112 (1:1000, rabbit, Fabgennix) or the PDE11A4-specific antibody PDE11A#1-8113A (1:10,000, chicken, Aves Labs custom). All the segment blots were probed with the PDE11A#1-8113A antibody only. The membranes were then incubated with a species-specific secondary antibody: anti-rabbit (111-035-144; 1:10,000; Jackson Immunoresearch, West Grove, PA, USA) and anti-chicken (Jackson Immunoresearch, 103-035-155; 1:40,000). The protein bands were visualized using the WesternSure Premium Chemiluminescent Substrate (#926-95000 LI-COR; Lincoln, NE, USA). Multiple film exposures were taken for each membrane to ensure the exposures fell within the linear range of the film. For the initial assessment of PDE11A presence in whole-eye tissue and eye segment tissue, all gels were run with the hippocampal controls. For the assessment of age-related changes in PDE11A4, the gels were run with the segment-specific KO controls (P28KO for retina; P28 and P500 KO for anterior and posterior—P500 shown). All PDE11A4 expression was normalized to Ponceau S staining intensity as a loading control, and densitometry was conducted on films scanned in at 1200 dpi using ImageJ software v1.48 (NIH).

### 2.5. Plasmid and Lentivirus Generation

As previously described [5,8,9], constructs were generated by Genscript (Piscataway, NJ, USA) that expressed either EmGFP alone containing an A206Y mutation to prevent EmGFP dimerization [51] or an EmGFP-tagged mouse *Pde11a4* (NM_001081033) with the *Bam*HI and *Xho*I recognition sequences added to the N-terminal and C-terminal, respectively. These constructs were initially generated on a pUC57 backbone and then subcloned into a pcDNA3.1+ mammalian expression vector using the *Bam*HI and *Xho*I sites (Life Technologies; Walthan, MA, USA). The QuickChange procedure/products were used to generate *Pde11a4* mutations as per manufacturer’s instructions (Agilent Technologies; Santa Clara, CA, USA). Oligonucleotide primers were generated by Integrated DNA Technologies (Coralville, IA, USA), and the mutated DNA sequences were verified by Functional Biosciences (Madison, WI, USA). mCherry in pcDNA3.1 was also generated by Genescript, and the TGN38 plasmid was purchased from Origene.

As previously described [5,9], constructs were subcloned into a proprietary “SPW” lentivirus transfer vector by the University of South Carolina Viral Vector Core using the restriction sites described above. The “SPW” backbone drives expression using the phosphoglycerate kinase 1 (PGK) promoter and is preferentially taken up by neurons even though it is a ubiquitous promoter in theory [5,9]. The lentiviruses delivered in 0.2 M sucrose/42 mM NaCl/0.84 mM KCl/2.5 mM Na_2_HPO_4_/0.46 mM KH_2_PO_4_/0.35 mM EDTA and the original titers were as follows: WT at 4.86 × 10^9^ TU/mL, Y727C at 2.23 × 10^9^ TU/mL, and M878V at 6.69 × 10^8^ TU/mL.

### 2.6. Cell Culture

As previously described [5,9], the HT22 cells were grown in Dulbecco’s Modified Eagle Medium (DMEM) with sodium pyruvate (GIBCO; Gaithersburg, MD, USA) supplemented with 10% filtered fetal bovine serum (R&D Systems, Bio-techne, Minneapolis, MN, USA) and 1% penicillin/streptomycin (GE Healthcare Life Sciences; Logan, UT, USA). The cells were passaged in T-75 flasks and incubated at 37 °C/5% CO_2_ to reach 65–75% confluency. The cells were lifted using TrypLE (GIBCO, Detroit, MI, USA) and then plated in supplemented DMEM. 1 μL of Lipofectamine 2000 (Invitrogen; Carlsbad, CA, USA) was used to transfect 0.375 μg of plasmid DNA per mL of Opti-MEM + GlutaMAX (+HEPES + 2.4 g/L Sodium Bicarbonate; GIBCO, Life Tech). After 7–16 h, Opti-MEM was replaced with supplemented DMEM, and the cells were left to continue growing until the time of fixation for microscopy or harvest for biochemical assays. Over the course of the experiments, the cells were sporadically tested for yeast, fungal, and bacterial infections (Invitrogen; Cat#:C7028), with negative results always obtained.

### 2.7. PDE Assay

PDE activity was measured using a coupled-enzyme assay that hydrolyzes [^3^H]cGMP or [^3^H]cAMP and then quantifies the resultant 5′-[^3^H]GMP and 5′-[^3^H]AMP that is isolated using low salt equilibrated anion-exchange columns (based on [52], with some modification as per [12]). To measure cAMP- and cGMP-PDE activity, the cell samples were sonicated in 20 mM Tris-HCl/10 mM MgCl_2_ in ultrapure water. 3ug of total protein/sample was then combined with a master mix that contained ~17,500–22,500 CPMS of ^3^H-cGMP or ^3^H-cAMP (Perkin Elmer; Waltham, MA USA) in 50 μL of 20 mM Tris-HCl/10 mM MgCl_2_ in ultrapure water. The samples were incubated at 37 °C for 10 min and then quenched with 0.1 M HCl and neutralized with 0.1 M Tris base (Trizma base, Sigma-Aldrich, Saint Louis, MO, USA). 37.5 µg of snake venom (V7000, Sigma-Aldrich) was added to each sample and incubated at 37 °C for 10 min. Each sample was then loaded into 5′ polystyrene chromatography columns with coarse filters (EvergreenSci; Rancho Dominguez, CA, USA) containing about 0.6 mL of settled DEAE Sephadex A-25 resin in 20 mM of Tris-HCl/0.1% sodium azide/0.5 M NaCl pH 6.80. The samples were then eluted 4× with 500 µL of 20 mM of Tris-HCl/0.1% sodium azide pH 6.80. The samples were then transferred to scintillation vials containing 4 mL of scintillation cocktail and counted using a Beckman liquid scintillation counter, with data expressed as CPMs/ug protein.

### 2.8. Ocular Physiology Studies

Ocular physiology studies were conducted using the Crispr B6/N *Pde11a* KO line. Ocular biometry was measured using spectral Domain Optical Coherence Tomography (SD-OCT Telesto, Thorlabs; Newton, NJ, USA), generating 3D images of the eye. The mice were positioned in front of the OCT in such a way that the pupil plane of the eye was close to perpendicular to the a-scan direction. The optical compartments were demarcated manually along the central axis using a custom graphical user interface written for MATLAB (version 9.1.0 (R2016b), The MathWorks Inc., Natick, MA, USA). The following boundary points were marked: outer and inner edge of the cornea, front and back edge of the lens, front and back edge of the retina corresponding approximately with the inner limiting membrane and ellipsoid zone, respectively. The measured axial compartment sizes (corneal thickness, anterior chamber depth, lens thickness, vitreous chamber depth, and retinal thickness) were converted to geometrical length values by dividing the optical path length by the approximate average refractive index (*n*) of the medium (*n* = 1.44 for the lens, *n* = 1.34 otherwise).

Refractive state was measured using the infrared eccentric photorefractor developed by Schaeffel et al. (software versions 2017 and 2019) [53,54]. The eye was oriented with the first Purkinje image in the center of the pupil, such that the estimated angle of the pupil plane was within 5 degrees of perpendicular to the camera axis. Refractive error was computed from the vertical gradient of the pupil reflection (10 samples per measurement, 99 measurements per eye). Measurements with the following criteria were included: minimal pupil size mean >1.70, maximum pupil size standard deviation < 0.05, pupil brightness 50–200, maximum pupil brightness standard deviation < 10, maximal refraction standard deviation < 2. We computed the mean and standard deviation from the 25–75% quartiles of the refraction data for each eye. In the case of an irregular reflection, generally caused by irregularity of the tear film, the measurement was postponed. The photorefractor setup was calibrated by measuring refractive errors after placing trial lenses (ranging from −10 to +20 D) at a fixed distance of 3 cm from the eye of an anesthetized mouse. We applied a correction for the magnification factor of the trial lenses (calibration factor software version 2017 = 1.08; version 2019 = 1.90).

Both ocular biometry and refractive state measurements were performed on both the right and left eye, as long as both were uninjured, under general anesthesia of ketamine/xylazine (KX: 80 mg/kg, 10 mg/kg body weight, resp.; Aneketin, Dechra veterinary products; Xylasan, Alfasan diergeneesmiddelen BV). Upon completion of the measurement, the mice were placed under a warming lamp and the eyes were moistened using artificial tears (Duodrops, Ceva sante animale BV). One hour after onset, the anesthesia was reversed using xylazine antagonist atipamezole (1 mg/kg body weight, Antisedan, Orion pharma) [55]. Two days later, the eyes were harvested and dissected for Western blot analyses, as described above.

### 2.9. Assessment of Cataracts

The cataract studies were conducted using the LacZ B6 *Pde11a* KO line. The mice were not anesthetized for the assessment of cataracts, as the experimenters were highly trained in mouse handling and sedation was not needed. The eyes of the subject mice were assessed using naked-eye visual inspection under the lighting of an animal-handling biosafety cabinet by experimenters blind to the genotype of the subject. The mice were restrained by hand, and both eyes were examined for dense clouding of the eye with varying opacities and coverage. If such cloudiness was observed, cataracts were noted as present.

### 2.10. Sleep Studies

Sleep studies were conducted as previously described [56] using the LacZ B6 *Pde11a* KO line. Under isoflurane anesthesia (induction 5%; maintenance 1–2%), adult mice were surgically implanted with EEG/EMG telemetry transmitters (HD-X02; Data Sciences International (DSI), St. Paul, MN, USA), adapted from previously published protocols [56,57]. Buprenorphine (0.1 mg/kg) and carprofen (5 mg/kg, s.c.) were administered for analgesic support at the start of the surgical procedure and as needed daily during post-operative recovery. Briefly, the head was secured in a stereotaxic frame (Stoelting Co., Wood Dale, IL, USA), the transmitter was inserted intraperitoneally through a dorsal abdominal incision, and an incision was made at the midline of the head. Two EMG leads were inserted and sutured into the dorsal neck muscle, and two EEG leads were anchored to surgical stainless-steel screws (P1 Technologies, Roanoke, VA, USA) placed into burr holes (0.5 mm diameter) at 1.9 mm anterior/+1.0 mm lateral and 3.4 mm posterior/−1.8 mm lateral relative to bregma and secured with dental cement (Stoelting Co., Wood Dale, IL, USA). The incisions were sutured, and the skin along the cement cap was reinforced with Gluture (Lambert Vet Supply, Fairbury, NE, USA). The animals were singly housed and recovered postoperatively for a minimum of 14 days prior to the start of sleep data acquisition.

Sleep data were acquired at a continuous sample rate of 500 Hz with Ponemah 6.10 software (DSI) in a designated room where the mice remained undisturbed. Data were averaged across a 2-day period for the males and a 4-day period for the females to span the estrous cycle. All digitized signal data were hand-scored by visual inspection with NeuroScore 3.0 (DSI) using 10 s epochs into one of three vigilance states: wake (low-amplitude, high-frequency EEG combined with high-amplitude EMG), NREM (high-amplitude, low-frequency EEG combined with low-amplitude EMG), or REM (low-amplitude, high-frequency EEG combined with very low EMG tone and muscle atonia [58,59]. The assessed parameters included relative onset, duration, and number of bouts of wake; non-rapid eye movement sleep (NREMS) and rapid eye movement sleep (REMS) sleep; and spectral power frequencies during REM sleep and NREM sleep, wherein discrete Fourier transform (DFT) estimated the EEG power spectrum for defined frequency bandwidths: delta (0.5–4 Hz), theta (4–8 Hz), alpha (8–12 Hz), sigma (12–16 Hz), and beta (16–20 Hz), as previously described [56].

### 2.11. Subcellular Localization

To study PDE11A4 subcellular localization using microscopy, cells in 24-well plates were transfected as described above and then fixed using 4% paraformaldehyde (Sigma-Aldrich) in 1× phosphate-buffered saline (PBS 10× Powder Concentrate, Fisher BioReagents) and stored in 1× PBS. The cells were then imaged using a Nikon Eclipse TE2000-E Inverted Microscope via a 10× objective equipped with a Photometrics CoolSNAP cf camera and a CoolLED pE-300lite LED illuminator. Representative images for each well were captured using MetaVue v6.2r6 software and were saved as jpeg files. As previously described [9], all images pertaining to an experiment were quantified by an experimenter blind to treatment using the same computer within the same position in the room, the same lighting conditions, and the same percent zoom. The images were loaded onto a gridded template to facilitate keeping track of the count locations within the image, and an experimenter scored each image box by box, with cells along the top and left edges of the entire image not included to follow stereological best practices. The images were quantified in a counterbalanced manner such that 1 picture from each condition was evaluated before moving onto a 2nd image from that condition. The experimenter classified the cells as exhibiting either cytosolic-only labeling, small puncta only (with/without cytosolic labeling), or large puncta (i.e., the size of a nucleus or bigger) with/without cytosolic labeling. Data were expressed as the % of the total number of labeled cells that exhibited small puncta only or large puncta.

### 2.12. Statistical Analysis

As previously described [5], data were collected by researchers blind to treatment, and the experiments were designed to counterbalance technical variables across the biological variables. For Western blots looking at age-related changes in PDE11A4, ImageJ (NIH) was used for collecting the densitometry data. For analyses of age-related changes in expression, data were expressed as a fold-change of P98 on each gel, in order to eliminate technical sources of variability blot to blot (e.g., differences in film exposures or batch concentrations). TData were analyzed by Sigmaplot 11.2 (San Jose, CA, USA), and Statistica v14 was used to conduct one-way, two-way (genotype × age), or three-way ANOVAs (genotype × sex × hour/frequency) when datasets passed tests of normality (Shapiro–Wilk test) and equal variance (Levene’s test). Repeated measures were used in the case of repeating measurements in the sleep data. Non-parametric ANOVA on Ranks were used if either normality or equal variance failed. Note that while males and females were included in the Western blot and ocular experiments, we were not sufficiently powered to formally analyze an effect of sex in these studies; however, we did visually inspect the data for a possible effect of sex. Following a significant main effect or interaction, post hoc analyses were performed using the LSD method (sleep studies), Student–Newman–Keuls, or Dunn’s method. Graphs were generated with GraphPad Prism 9 software. Data are plotted as mean ± SEM, with individual data points shown overtop (circles = females; squares = males). Significance is defined as *p* < 0.05.

## 3. Results

### 3.1. Western Blots Identify PDE11A4 Protein in Whole-Eye Tissue and Individual Eye Segments

To determine if PDE11A protein is present in ocular tissue, Western blots were conducted using whole eyes from WT and KO male and female mice from three *Pde11a* mouse lines (i.e., Crispr C57BL/6N, LacZ BALB/cJ, LacZ C57BL/6J). Whole eyes were processed using either the ice-cold Buffer #1 or the boiling Buffer #2 to ensure reliability of detection across varied preparations. The samples were then probed with either a pan-PDE11A antibody capable of detecting all four PDE11A isoforms or a PDE11A4-specific antibody. *Pde11a* WT and KO hippocampi were loaded as positive and negative controls, respectively, given the well-established expression of PDE11A4 protein in this brain region [3,4,5,49]. The pan-PDE11A antibody reliably detects PDE11A4 but not PDE11A1-3 in both hippocampus and whole eyes of *Pde11* WT mice but not KO mice (Figure 1A). The presence of PDE11A4 in these tissues was confirmed with the PDE11A4-specific antibody (Figure 1B). We then determined if PDE11A4 expression was enriched in one segment of the eye over another. The retina (Figure 1C), anterior segment (i.e., cornea, ciliary body and muscle, and zonules; Figure 1D), and posterior segment (i.e., sclera, choroid, Bruch’s membrane, retinal pigmented epithelium, and parts of the optic nerve; Figure 1E) from the Crispr B6/N line of *Pde11a* WT and KO mice were dissected from male and female mice and show robust expression of PDE11A4 protein in the retina (Figure 1C), with much less expression in the anterior (Figure 1D) and posterior segments (Figure 1E).

### 3.2. PDE11A4 Expression in the Eye Changes with Age in A Segment-Specific Manner

Given that PDE11A4 protein expression increases across the lifespan in the hippocampus [3,5,49], we next determined if PDE11A4 protein expression differed in the eye as a function of age (i.e., P7–P500). WT samples were analyzed at P7 (in the retina only, as the anterior and posterior samples were insufficiently concentrated at this age; n = 3/sex), P28, P56, P98, P200, and P500 (n = 4/sex/age P28–P500) with a segment-specific KO sample loaded as a negative control. In the retina (Figure 2A), PDE11A4 protein expression decreases from P7 to P56–P500 between the sexes (ANOVA on Ranks: H(5) = 15.18, *p* = 0.01; post hoc vs. P7: P56 *p* = 0.02, P98 *p* = 0.0328, P200 *p* = 0.0368, P500 *p* = 0.0368). Similarly, PDE11A4 protein expression in the posterior segment decreases from P28 to P500 (Figure 2C; ANOVA on Ranks: H(4) = 17.73, *p* = 0.0014; post hoc vs. P500 for each group: *p* = 0.049–0.0022). In contrast, however, PDE11A4 protein expression in the anterior segment increases between P28 and P500 across both sexes (Figure 2B; ANOVA on Ranks: H(4) = 9.84, *p* = 0.043; post hoc vs. P500 for each group: *p* ≤ 0.049). Although the current study was not powered to analyze for the effect of sex, a visual inspection of the data suggests that the age-related decrease in PDE11A4 in the posterior segment is more pronounced in females than males. Together, these data suggest that ocular PDE11A4 expression changes with age in a segment-specific manner.

### 3.3. PDE11A4-Y727C and -M878V Variants Exhibit Reduced cAMP-PDE and cGMP-PDE Activity in Neural Cells

Given that PDE11A4 is enriched in neural cells, we next determined if the PDE11A4-Y727C variant exhibits impaired catalytic activity relative to PDE11A4-WT in a neural cell line (i.e., HT22 cells). Effects of the nearby PDE11A4-M878V variant, previously associated with high-risk suicide [48] but not myopia nor sleep [14,16], were also measured to determine the specificity/selectivity of the PDE11A4-Y727C effects. In HT22 cells (n = 6 biological replicates/group), Y727C completely eliminates cAMP hydrolytic capability (Figure 3A; F(3,20) = 15.78, *p* < 0.0001; post hoc vs. Y727C: WT *p* = 0.0002, GFP *p* = 0.7785), whereas M878V only reduces cAMP hydrolysis by ~35% (post hoc vs. M878V: WT *p* = 0.04479, GFP *p* = 0.00041, Y727C *p* = 0.003). Similarly, Y727C reduces mPDE11A4-mediated cGMP hydrolysis by almost 70% (Figure 3B; F(3,20) = 63.85, *p* < 0.0001; Post hoc vs. Y727C: WT *p* = 0.0001, GFP *p* = 0.0009), whereas M878V only reduces cGMP hydrolysis by 20% (post hoc vs. M878V: WT *p* = 0.0248, GFP *p* = 0.0001). These differential effects (i.e., Y727C effect > M878V effect and cAMP effect > cGMP effect) were replicated in a second set of experiments with different biological replicates (Appendix A).

### 3.4. Loss of PDE11A Does Not Alter Eye Growth

Given that the Y727C variant loses all cAMP-hydrolytic activity and 70% of its cGMP hydrolytic activity, we next determined if genetic deletion of *Pde11a* would affect eye growth using the Crispr B6/N *Pde11a* mouse line. To assess the effect of *Pde11a* deletion on eye growth, we measured ocular biometry and refractive state from postnatal day (P) 28 through P500. As previously described for the LacZ B6 *Pde11a* KO mice [4], the Crispr B6/N *Pde11a* KO mice were generally noted as healthy relative to the WT mice throughout the experiments, with no significant effect of genotype on age-related increases in body weight (failed equal variance; ANOVA on Ranks for effect of age: H(4) = 49.98, *p* < 0.0001; Figure 4A). Axial length (Figure 4C; effect of age: F(4,113) = 463.02, *p* < 0.0001), anterior chamber depth (Appendix A; effect of age: F(4,113) = 380.84, *p* < 0.0001), and lens thickness (Appendix A; failed equal variance; ANOVA on Ranks for effect of age: H(4) = 121.82, *p* < 0.0001) increase with age; conversely, vitreous chamber depth shrinks (Appendix A; failed normality; ANOVA on Ranks for effect of age: H(4) = 110.12, *p* < 0.0001) and retinal thickness does not significantly change with age (Figure 4B). There are no significant effects of genotype on any of these measures, suggesting normal eye growth in the Crispr B6/N *Pde11a* KO mice relative to WT mice. At P28, the eyes from *Pde11a* KO mice exhibit a significantly different refractive error relative to *Pde11a* WT eyes (Figure 4D; effect of age x genotype: F(4,104) = 2.81, *p* = 0.029; post hoc: P28 WT versus KO, *p* = 0.0122), but the *Pde11a* KO eyes perform nearly identical to those of the WT mice at P56 (post hoc: KO-P28 versus KO-P56, *p* = 0.0211; P56 WT versus KO, *p* = 0.9997).

Given that we observed age-related changes in PDE11A4 protein expression in the eye, we also assessed the prevalence of cataracts in three cohorts of male and female LacZ B6/J *Pde11a* mice spanning 12–18 months of age. There was no difference in the prevalence of cataracts between the genotypes, with 4/92 WT and 7/87 KO mice exhibiting cataracts (*p* = 0.3185).

### 3.5. Pde11a Deletion Minimally Improves Sleep Quality

Given the link between PDE11A4-Y727C variant and increased sleep duration and efficiency [16], coupled with the fact that PDE11A4 protein is found in brain regions critical for sleep (i.e., the hippocampus and hypothalamus [5]), we next determined if the duration, number of bouts, or onset of wake, NREM, and/or REM differ between male and female LacZ B6/J *Pde11a* KO mice relative to WT same-sex littermates. When all epochs are analyzed in a three-way RM ANOVA for genotype, sex, and hour, the LacZ B6/J *Pde11a* KO mice do not significantly differ from WT littermates in terms of the total duration (Figure 5) or number of bouts (Appendix A) of wake, NREM, or REM during any time across a 24 h cycle. That said, an independent analysis of the REM duration during the first hour of the light cycle does show that male LacZ B6/J *Pde11a* KO mice spend longer in REM relative to male WT littermates (Figure 5F; effect of genotype x sex: F(1,18) = 5.33, *p* = 0.0331; post hoc: male KO vs. male WT, *p* = 0.0027). A similar trend is observed in males in terms of the number of REM bouts during the first hour (Appendix A; effect of genotype × sex: F(1,18) = 3.52, *p* = 0.0771). The LacZ B6/J *Pde11a* KO mice show a similar onset to NREM as the WT littermates (Figure 5E) but exhibit a significantly shorter onset to REM (Figure 5H; effect of genotype: F(1,18) = 6.01, *p* = 0.0246). Although the effect of genotype × sex does not rise to the level of significance, a visual inspection of the data suggests the effect of *Pde11a* deletion on REM onset is driven by the males.

Next, we analyzed the delta power during NREM and the theta power during REM. Both male and female LacZ B6/J *Pde11a* KO mice significantly differ from the same-sex WT littermates at the lower frequencies within the delta range, albeit in opposite directions (Figure 5I,J; three-way RM ANOVA for effect of genotype × sex × frequency: F(7,119) = 7.34, *p* < 0.0001). Post hoc analyses reveal that male LacZ B6/J *Pde11a* KO mice showed a significant reduction in power relative to WT males at 0.5 Hz (*p* = 0.004), 1 Hz (*p* = 0.0026), and 1.5 Hz (*p* = 0.0122), whereas the female LacZ B6/J *Pde11a* KO mice show a significant increase in power relative to the WT females at 0.5 Hz (*p* = 0.006), 1 Hz (*p* = 0.0017), and 1.5 Hz (*p* = 0.0263). A strong trend towards an opposing genotype effect in male versus female LacZ B6/J *Pde11a* KO mice is also observed across the higher frequencies of theta power during REM (Figure 5K,L; three-way RM ANOVA for effect of genotype × sex: F(1,136) = 4.41, *p* = 0.051). Taken together, these results suggest the impact of *Pde11a* deletion on unperturbed sleep is subtle and sex-specific.

### 3.6. PDE11A4-Y727C and -M878V Variants Dramatically Alter the Subcellular Compartmentalization of the Enzyme

Given the minimal effect that *Pde11a* deletion had on eye growth and sleep–wake measurements, coupled with the fact that the location of a PDE is just as important to its overall function as is its catalytic activity [1], we next determined if the Y727C mutation changes the subcellular compartmentalization of PDE11A4. The impact of M878V on subcellular compartmentalization was again measured as a comparator. Select studies were performed with human PDE11A4 (hPDE11A4), but most were conducted with mouse PDE11A4 (mPDE11A4) since we are using a mouse HT22 cell line and injections in mouse brain to explore functional significance. Immunocytochemistry of untagged human PDE11A4 (hPDE11A4) reveals that both Y727C and M878V dramatically change the subcellular compartmentalization of hPDE11A4 in both HT22 cells (Figure 6A) and COS-1 cells (Appendix A). Whereas wild-type hPDE11A4 (hPDE11A4-WT) exhibits both a diffuse localization throughout the cytoplasm and points of punctate accumulation (as previously described [5,8,9]), the hPDE11A4-Y727C variant shows only the diffuse localization throughout the cytoplasm. The hPDE11A4-M878V variant similarly reduces the punctate accumulation of the enzyme.

We then determined if this phenotype was conserved across species by testing mouse PDE11A4 (mPDE11A4). We also examined the extent to which any compartmentalization effect may differ between Y727C and M878V versus a phosphomimic mutation at serine 162 (S162D) previously reported to trigger this type of diffuse localization of PDE11A4 [5]. Finally, we ascertained if there might be a differential effect on small PDE11A4 accumulations versus larger accumulations. Indeed, the dispersing effect of the Y727C and M878V variants occurs with mPDE11A both in cell culture (Figure 6B and Appendix A) and in the brain (Appendix A). Y727C reduces the presence of smaller punctate accumulations of mPDE11A4 to the same extent as does mPDE11A4-S162D in HT22 cells (Figure 6C; n = 18/group; combined analysis of three experiments with similar results: F(2,51) = 40.76, *p* < 0.0001; post hoc each group vs. WT: *p* = 0.0001), but to a greater extent than S162D in COS-1 cells (Appendix A). This reduction in small PDE11A4 puncta appears in tandem with Y727C increasing the presence of larger mPDE11A4 accumulations in both HT22 cells (Figure 6E; failed normality and equal variance; ANOVA on Ranks H(2) = 18.88, *p* < 0.0001; post hoc WT vs. Y727C *p* < 0.0001) and COS-1 cells (Appendix A). M878V also reduces the presence of smaller punctate accumulations of PDE11A4 to a similar extent, as does S162D in both HT22 (Figure 6D; H(2) = 33.36, *p* < 0.0001; post hoc vs. M878V: WT *p* < 0.0001, S162D *p* = 1.0) and COS-1 cells (Appendix A). However, M878V reduces the presence of smaller PDE11A puncta in HT22 cells without changing the occurrence of larger PDE11A accumulations in HT22 cells (Figure 6F; H(2) = 2.94, *p* = 0.23).

### 3.7. The Y727C Variant Dictates Localization of mPDE11A4-WT When Co-Expressed

Given that patients are typically heterozygous for these mutations in PDE11A4 [13,15,48,60,61,62], coupled with the fact that PDE11A4 exists as a homodimer [2], we next determined the effect of expressing a 50%–50% mix of the mutant and WT cDNA as opposed to 100% WT or 100% mutant (note: all groups had the same total amount of plasmid DNA transfected). Surprisingly, when the mY727C variant is co-expressed with mPDE11A4-WT in HT22 cells, the phenotype of the mY727C variant completely dominates, with mY727C+WT being just as dispersed as Y727C alone (Figure 6H; H(4) = 53.90, *p* < 0.0001; post hoc: WT vs. Y727C *p* < 0.0001, WT vs. Y727C+WT *p* < 0.0001, Y727C vs. Y727C+WT *p* = 1.0). The same effect was observed in the COS-1 cells (Appendix A). In contrast, mM878V co-expressed with mPDE11A4-WT results in the mathematically predicted 50% effect size of M878V alone in HT22 cells (Figure 6H; post hoc: WT vs. M878V *p* < 0.0001, WT vs. M878V+WT *p* = 0.187, M878V vs. M878V+WT *p* = 0.01) and COS-1 cells (Appendix A). This suggests that mPDE11A4-Y727C readily forms heterodimers with PDE11A4-WT, resulting in almost all the dimers containing the variant, whereas the M878V variant forms homodimers with itself to produce 50% variant homodimers and 50% WT homodimers.

### 3.8. The Effects of the Y727C and M878V Variants on PDE11A4 Subcellular Compartmentalization Do Not Require Phosphorylation of S162 nor Dephosphorylation of S117/S124

Next, we explored the potential mechanism by which the Y727C and M878V variants dramatically alter PDE11A4 subcellular localization. In addition to phosphorylation at S162, dephosphorylation at S117/S124 can also reduce the punctate accumulation of both human and mouse PDE11A4 [5]. As such, we compared the levels of PDE11A4-pS117/pS124 between the WT and variant versions of hPDE11A4 and mPDE11A4 in HT22 cells. Consistent with the fact that preventing phosphorylation of S117/S124 disperses PDE11A4 [5], hPDE11A4-Y727C (0.38 ± 0.05 arbitrary units (A.U.)) exhibits ~63% less pS117/pS124 than hPDE11A4-WT (1.00 ± 0.01 A.U.; F(2,15) = 9.37, *p* = 0.0022; post hoc: hWT vs. hY727C, *p* = 0.0024); however, hPDE11A4-M878V shows no such loss of phosphorylation (1.03 ± 0.18; post hoc: hWT vs. hM878V, *p* = 0.871; n = 8/plasmid). Similarly, mPDE11A4-Y727C shows ~51% less pS117/pS124 than mPDE11A4-WT (Figure 6I; Rank Sum test: T(10,10) = 143, *p* = 0.0046) with no reduction in phosphorylation by mM878V relative to mWT (Figure 6J; t(14) = 0.71, *p* = 0.4895). We next determined if the dispersing effects of the Y727C or M878V mutations required the dephosphorylation of S117/S124 or the phosphorylation of S162. Surprisingly, neither mechanism proves necessary for the dispersal of small PDE11A4 puncta because neither the phosphomimic mutations S117D/S124D nor the phosphoresistant mutation S162A diminish the ability of Y727C to reduce small mPDE11A4 puncta (Figure 6L; equal variance failed, ANOVA on Ranks: H(3) = 27.32, *p* < 0.0001; post hoc vs. WT: Y727C *p* = 0.0006, Y727C/S162A *p* < 0.0001, Y727C/S117D/S124D *p* = 0.0002; post hoc vs. Y727C: Y727C/S162A *p* = 1.0, Y727C/S117D/S124D *p* = 1.0). Similarly, neither mechanism is required for Y727C to increase large mPDE11A4 punctate accumulations (Figure 6M; ANOVA on Ranks: H(3) = 14.57, *p* = 0.0022; post hoc vs. Y727C: WT *p* = 0.028, Y727C/S162A *p* = 1.0, Y727C/S117D/S124D *p* = 1.0). Phosphorylation of S162 is also not needed for the dispersal effects of M878V; however, M878V is not able to fully block the pro-accumulating effect of S117D/S124D as did Y727C (Figure 6N; F(3,44) = 163.62, *p* < 0.0001; post hoc vs. WT: M878V *p* = 0.0001, M878V/S162A *p* = 0.0002, M878V/S117D/S124D *p* = 0.0001; post hoc vs. M878V: M878V/S162A *p* = 0.182, M878V/S117D/S124D *p* = 0.042).

### 3.9. The Effects of Y727C on PDE11A4 Subcellular Compartmentalization Are Due to the Loss of the Tyrosine Impacting Processing via the Trans-Golgi Network

Given that a tyrosine (Y) can be phosphorylated, we next asked if the Y727C variant disperses PDE11A4 due to the general loss of the Y, the specific loss of phosphorylating the Y, or the specific gain of the cysteine. Relative to Y727C, a Y727A phosphoresistant mutant and a Y727D phosphomimic mutant equivalently reduce the presence of small mPDE11A4 puncta (Figure 6P; ANOVA on Ranks: H(3) = 27.32, *p* < 0.0001) and increase the presence of large mPDE11A4 punctate accumulations (Figure 6Q; F(3,44) = 7.89, *p* = 0.0002). This suggests it is neither the loss of pY727 nor the specific gain of the cysteine that drives the effects of Y727C on subcellular compartmentalization, but rather the loss of the tyrosine itself that may alter intramolecular interactions required for proper protein conformation [63].

Previous work in the lab [5] suggested that PDE11A4 is packaged/repackaged via the trans-Golgi network, a major sorting station for secretory and membrane proteins. Further, the larger PDE11A4 punctate accumulations appear to partially co-localize with the Golgi apparatus (Appendix A). As such, we determined if the Y727C variant impacts mPDE11A4 processing through the trans-Golgi network to elicit its unique effect of increasing the presence of large PDE11A4 accumulations. To do so, we overexpressed either mCherry alone as a negative control or a red fluorescent protein-tagged trans-Golgi network 38 protein (RFP-TGN38, also known as TGOLN, the orthologue to human TGOLN2). TGN38 is a type I integral membrane protein that cycles between early endosomes and the trans-Golgi network to regulate exocytic vesicle formation [64]. Consistent with previous evidence suggesting PDE11A4 is processed via the Golgi [5], overexpressing TGN38 increases the presence of small mPDE11A4-WT, mPDE11A4-Y727C, and mPDE11A4-M878V puncta (Figure 6R; effect of mCherry plasmid: F(1,14) = 28.77, *p* = 0.001). In contrast, overexpressing TGN38 only increases the presence of large mPDE11A4-Y727C puncta, amplifying the effect of the variant on this measure two-fold (Figure 6S; interaction effect: F(2,14) = 5.82, *p* = 0.0145; post hoc vs. mCherry alone: WT+TGN38 *p* = 0.194, Y727C+TGN38 *p* = 0.0002, M878V+TGN38 *p* = 0.122).

## 4. Discussion

In this study, we demonstrate that the PDE11A-Y727C and PDE11A-M878V variants significantly reduce PDE11A4 catalytic activity and dramatically alter its intracellular trafficking. Interestingly, the effects of the Y727C variant that are linked to reduced myopia risk and increased sleep quality [14,16] are quite distinct from that of the nearby M878V variant that is linked to high-risk suicide [48] but not myopia or sleep [14,16]. Importantly, we show for the first time that PDE11A4 protein is expressed in the eye, with a particular enrichment in the retina (Figure 1 and Figure 2), putting PDE11A4 in a position to regulate not only brain functions that might impact sleep quality, but also eye physiology that might influence susceptibility to myopia. Indeed, the PDE11A substrates cAMP and cGMP are known to regulate the physiological function of the normal [65,66,67,68,69,70,71,72] and myopic eye [73,74,75,76], as well as sleep [19]. Even though the Y727C mutation significantly reduces PDE11A4 catalytic activity (Figure 3 and Appendix A), genetic deletion of *Pde11a* in mice does not affect eye growth (Figure 4B,C and Appendix A) and only minimally affects refractive error (Figure 4D) and sleep quality (Figure 5 and Appendix A). Therefore, we propose that the protective impact of the Y727C mutation in humans may reflect a gain of function, in which PDE11A4 displaces another PDE from its pool of cyclic nucleotides, leaving them undegraded. Indeed, a wide array of disorders are caused by gain-of-function germline mutations in other PDE families [77]. That said, it should be noted that we only examined baseline measures herein. It is quite possible that in the context of a disease-like state (e.g., a genetic mouse model with increased susceptibility to myopia, a model of sleep deprivation, etc.), *Pde11a* deletion would provide a protective action.

### 4.1. PDE11A4 Expression Is Greatly Enriched in the Retina vs. Anterior or Posterior Segments of the Eye

This study is the first to report that PDE11A4 protein—not PDE11A1, A2, or A3—is expressed in the eye (Figure 1A). From P7 to P500, ocular PDE11A4 expression is enriched in the retina with minimal expression in the anterior and posterior segments (Figure 1C–E). These findings support previous research showing an enrichment of PDE11A4 in neural tissue, including the hippocampus of the brain, the spinal cord, and the dorsal root ganglia [2,5]. Interestingly, compartment-specific expression within a given tissue appears to be the modus operandi for PDE11A4. Within the hippocampus, this enzyme is only expressed in the CA1 and subiculum subfields and not in CA3 or the dentate gyrus. Further, PDE11A4 levels are significantly higher in the ventral versus the dorsal hippocampus [3,4,8,49]. Outside of the brain, PDE11A4 has been detected in the dorsal root ganglia and the spinal cord [78], where its expression is limited to somatostatin-expressing interneurons of the dorsal horn [79]. In addition to the nervous system, PDE11A4 has been observed in the adrenal gland of humans [80], but not mice [78], with hPDE11A4 present in the adrenal cortex and nodules but not the adrenal medulla region [81]. Even within the retina, PDE11A4 protein expression is likely to be parsed because an RNA sequencing study of human retina showed PDE11A mRNA expressed in only a subtype of cone ON-bipolar cells, horizontal cells, GABAergic amacrine cells (VIP+ and starburst), retinal ganglion cells, and Muller glia and not at all in photoreceptors [25].

### 4.2. PDE11A4 Expression in the Eye Changes with Age in a Segment-Specific Manner

The observed age-related changes in PDE11A4 protein expression are not consistent across eye segments. This study reveals a significant decline in retinal PDE11A4 levels after P7, after which, PDE11A4 levels remain relatively stable up to P500 (Figure 2A). Interestingly, this sharp reduction in retinal PDE11A4 expression coincides with the approximate period when mice open their eyes for the first time (P10–13). This correlation is particularly interesting given a report showing that light exposure can regulate PDE11A4 expression in vitro [32]. In contrast, we observed a gradual age-related increase in PDE11A4 expression in the anterior segment (Figure 2B), which would reduce cAMP and cGMP. In the anterior portion of the eye, cAMP normally acts to stimulate fluid transport across the ciliary epithelium [65] and can prevent TNF-α-induced barrier dysfunction at the corneal endothelium [82]. In contrast, the cGMP signaling pathway facilitates sustained relaxation of the trabecular meshwork [83], which regulates the outflow of aqueous humor and regulates intraocular pressure (IOP) [84]. Elevated IOP, which is more common in older age [85], can lead to the development of glaucoma [84] and high myopia-associated optic neuropathy [86]. Importantly, metabolic markers also exhibit a segment-specific age-related change in the eye [87]. Compartment-specific age-related changes in PDE11A4 expression, phosphorylation, and trafficking have also been described in the hippocampus [5]. Importantly, preventing or reversing these compartment-specific age-related increases in hippocampal PDE11A4 expression and accumulation rescues age-related cognitive decline of social associative memories [5,9]. Thus, it will be important for future studies to better understand how PDE11A4 function may change within an eye-segment as a function of age.

### 4.3. The PDE11A4-Y727C and -M878V Variants Impair cAMP Hydrolysis More So than cGMP Hydrolysis

It is not surprising that the PDE11A4-Y727C and -M878V variants reduce both the cAMP and cGMP hydrolytic activity of PDE11A4 (Figure 3 and Appendix A), given that they are located within the C-terminal catalytic domain [6,78]. What is surprising, however, is the fact that both the Y727C and M878V variants impair PDE11A4 cAMP-hydrolytic activity more so than cGMP-hydrolytic activity. This is reminiscent of the fact that PDE5 inhibitors with PDE11A off-target activity are capable of inhibiting PDE11A cGMP-hydrolytic activity at much lower IC50s than cAMP-hydrolytic activity (c.f. [6,78]). A detailed kinetic analysis of purified PDE11A4 versus PDE11A4-Y727C and PDE11A4-M878V will be of interest to future studies, including the determination of the turnover number and K_m_ constant. Together, these results suggest it may be possible to therapeutically target one PDE11A4 catalytic activity more so than the other, providing yet another degree of freedom in selectively targeting cyclic nucleotide signaling in a tissue-specific and subcellular compartment-specific manner.

### 4.4. PDE11A4-Y727C May Reflect a Gain-of-Function Mutation

Even though PDE11A4-Y727C dramatically reduces both cAMP- and cGMP-catalytic activity, genetic deletion of *Pde11a* does not change eye growth (Figure 4 and Appendix A) and only minimally improves sleep quality (Figure 5 and Appendix A). When working with constitutive KO mice, compensation elsewhere in the signaling cascade is always a possibility. That said, at least in the hippocampus, genetic deletion of *Pde11a* does not appear to trigger such compensation and results in increased lithium responsiveness, stronger remote social memory formation, and changes in social preferences [3,4,5,7,8,11,12]. In addition to impairing catalytic activity, however, the Y727C variant also dramatically alters PDE11A4 subcellular compartmentalization (Figure 6 and Appendix A). In contrast to the punctate accumulation observed with PDE11A4-WT, a largely diffuse localization throughout the cytoplasm is exhibited by PDE11A4-Y727C and -M878V (Figure 6A,B and Appendix A). More interestingly, the Y727C variant appears to homodimerize with PDE11A4-WT and dominates the function of the dimerized protein (Figure 6G,H and Appendix A). Thus, the Y727C variant may also prove protective in the context of age-related cognitive decline since increased expression and clustering of PDE11A4 protein is associated with age-related cognitive decline of social memories [5,9].

The effects of the Y727C and M878V variants on PDE11A4 subcellular compartmentalization do not appear to be related to the PDE11A4 phosphorylation signatures previously established to disperse PDE11A4 accumulation [5]. Despite reducing PDE11A4-pS117/pS124, the Y727C variant triggers the largely diffuse localization of PDE11A4 independently of dephosphorylating PDE11A4-S117/S124 or phosphorylating PDE11A4-S162 (Figure 6L,M). This is also true for the M878V variant (Figure 6N,O). Further, mutating Y727 to a phosphoresistant alanine (A) or a phosphomimic aspartate (D) also reduces the accumulation of small PDE11A4 puncta. If it had been due to the loss of pY727, then the Y727D mutant would have increased accumulation relative to Y727A. This is highly interesting given that age-related increases in hippocampal PDE11A4-pS117/pS124 that promote PDE11A4 accumulation are associated with cognitive decline in mice [5]. The fact that both Y727A and Y727D had the same effect on PDE11A4 subcellular compartmentalization as Y727C also suggests the variant phenotype is not specifically related to gaining a cysteine at this position. Rather, the effect of the Y727C mutation appears to be due to the loss of the tyrosine itself at this position, a residue often implicated in critical intramolecular interactions required for proper protein conformation [63].

The ability of Y727C to reduce the presence of small PDE11A4 puncta while increasing the presence of large PDE11A4 puncta may be related to altered Golgi processing of the mutated protein. Although PDE11A4-Y727C, -M878V, and -S162D all equivalently reduce the presence of small PDE11A4 puncta (Figure 6C,D), only the Y727C variant triggers a significant increase in large PDE11A4 accumulations in HT22 cells (Figure 6E,F). Previous work from our lab suggests that PDE11A4 protein is packaged/repackaged via the trans-Golgi network [5], and we find here that the large PDE11A4 accumulations can partially co-localize with the Golgi apparatus (Appendix A). Further, overexpression of TGN38, a type I integral membrane protein that cycles between early endosomes and the trans-Golgi network to regulate exocytic vesicle formation [64], increases the presence of small PDE11A4 puncta of PDE11A4-WT, -Y727C, and -M878V. In contrast, overexpression of TGN38 increases large accumulations of PDE11A4-Y727C only, doubling the effect size of the variant (Figure 6R,S). Interestingly, neurons are particularly vulnerable to Golgi stress [88]. Given that the functional impact of the Y727C variant is due to the loss of the tyrosine (Figure 6P,Q), we hypothesize that the Y727C variant loses a critical intramolecular interaction required for proper conformation [63], which ends up affecting packaging/repacking of the protein via the trans-Golgi network (Figure 6R,S).

Given the minimal effect of *Pde11a* deletion on ocular biometry and sleep, coupled with the strong influence of the Y727C variant on the subcellular compartmentalization of PDE11A4, we suggest the net effect of this variant is actually a gain-of-function. Such an effect could be produced by altering the overall function of the trans-Golgi network as described above, or by introducing the residual PDE11A4-Y727C cGMP-hydrolytic activity to a pool that would not otherwise be degraded. Alternatively, such an effect could be elicited if this nearly catalytically dead variant displaced another PDE from its respective pools of cyclic nucleotides, thereby increasing cyclic nucleotide levels elsewhere. Indeed, several members of the PDE family have been identified in eye tissues [89,90,91,92,93,94]. In rats, PDE1B is uniformly expressed across the retina, while PDE1A is present in the outer retina and PDE1C is preferentially expressed in the inner nuclear layer [89]. PDE2, PDE5, and PDE9 contribute to cGMP metabolism in retinal pigment epithelium cells in rats [90]. PDE5 also plays a role in the regulation of ocular blood flow, given its expression in the retinal and choroid vasculature, as well as in ganglion and bipolar cells [91]. PDE6, expressed in retinal photoreceptors [91] is an integral part of the phototransduction cascade where it hydrolyzes cGMP signals in response to light stimulation [95]. PDE7B and PDE8A have also been reported in the eye, although their contribution to eye function is not currently known [93,94]. Thus, it is of high interest to drive expression of PDE11A4-Y727C in a region-specific manner (e.g., retina versus ventral hippocampus), perhaps even in a cell type-specific manner within a region, to better understand how this potential gain-of-function impacts both normal and disturbed eye physiology and sleep.

## 5. Conclusions

In this study, we show for the first time that PDE11A4 protein is present in ocular tissue and propose that the Y727C variant is a gain-of-function mutation, by virtue of altering its subcellular compartmentalization. These results support previous work establishing a key role of PDE11A4 in hippocampal-mediated brain function and further suggest a role for PDE11A4 in regulating eye function and reducing myopia risk. It will be important for future studies to fully understand the physiological impact of PDE11A4-Y727C in vivo and to determine if PDE11A4 might be harnessed for therapeutic gain in the context of myopia, sleep dysregulation, or age-related cognitive decline.

## Figures and Tables

**Figure 1 cells-12-02839-f001:**
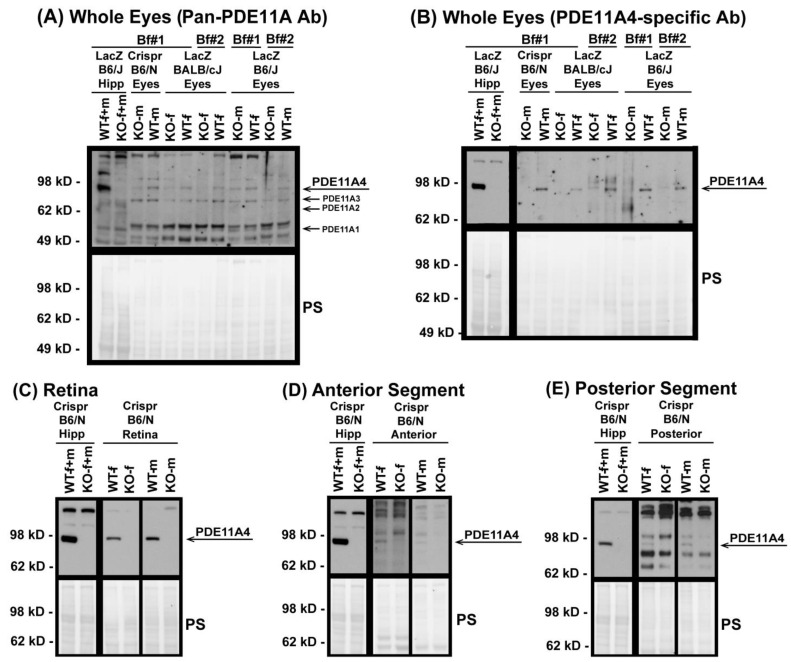
PDE11A4 protein is expressed in the mouse eye, with enrichment in the retina. Using samples from multiple *Pde11a* mouse lines processed using multiple lysis buffers, PDE11A4 protein was reliably detected in whole eyes from WT mice (but not KO mice) by (**A**) a pan-PDE11A antibody (A4~97 kD; A3~78 kD; A2~66 kD; A1~56 kD), as well as a (**B**) PDE11A4-specific antibody. To determine potential enrichment in one segment over another, eyes from the Crispr B6/N line were dissected into retina, anterior segment, and poster segment and were lysed in buffer #1 and then probed with the PDE11A4-specific antibody. (**C**) Retina shows robust PDE11A4 protein expression requiring minutes-long film exposures to visualize. In contrast, (**D**) anterior segment and (**E**) posterior segment show minimal expression that requires overnight exposures to visualize. Note: PDE11A4 expression in the hippocampus (Hipp) is much higher than that in the eye, requiring far shorter film exposures to reveal strong specific signals in WT; hence, those exposures are shown blocked off from the eye samples in all but panel A. Brightness and contrast of images adjusted for graphical clarity. Ab—antibody, Bf#1—ice-cold buffer #1, Bf#2—boiling buffer #2, WT—wild-type, KO—knockout, f—female, m—male, PS—Ponceau S stain.

**Figure 2 cells-12-02839-f002:**
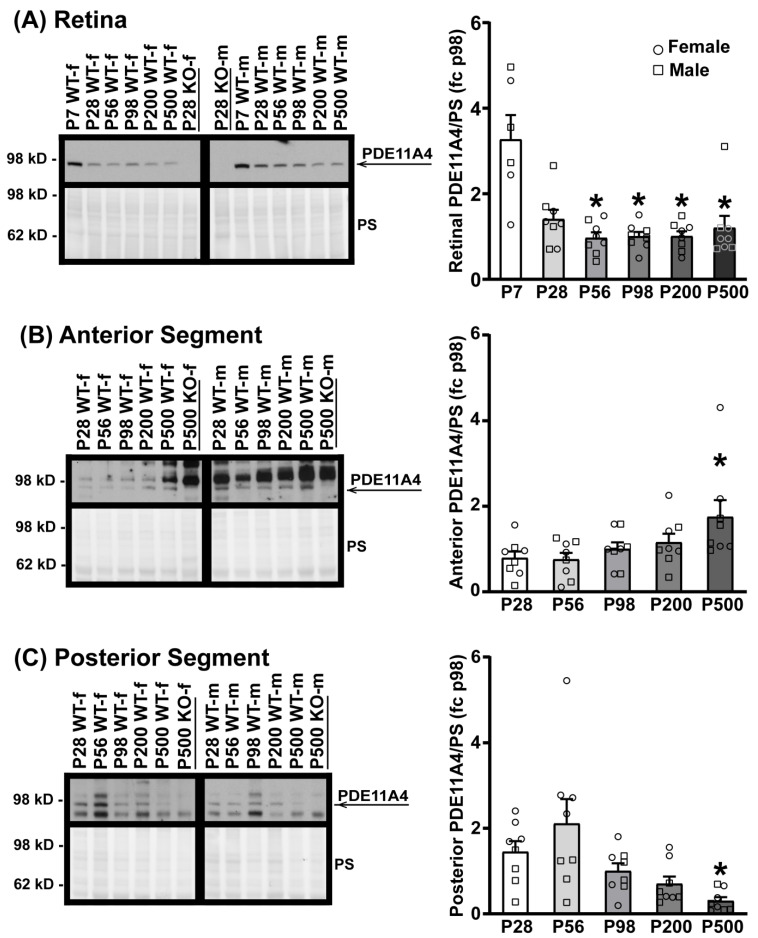
PDE11A4 protein expression in the eye changes with age. Eye segments from Crispr B6/N *Pde11a* WT and KO male and female mice were homogenized using buffer #1 and probed for PDE11A4 with the cleaner PDE11A4-specific antibody. Expression levels were analyzed at P7 (in retina only; n = 3/sex), P28, P56, P98, P200, and P500 (n = 4/sex/age P28–P500) and expressed as a fold-change (fc) of P98. (**A**) In the retina, PDE11A4 expression is highest at P7 in both sexes, after which, expression levels decrease and stay relatively stable from P28 to P500. (**B**) In the anterior segment, PDE11A4 expression increases with age in both sexes and peaks at P500. (**C**) In the posterior segment, PDE11A4 expression decreases with age from P28 to P500. Visual inspection of the data suggests the effect in the posterior segment is greater in females than in males. * vs. lowest age, *p* = 0.049–0.0022. Data expressed as mean ± SEM. Brightness and contrast of images adjusted for graphical clarity. WT—wild-type, KO—knockout, PS—Ponceau S stain, f—female (circles), m—male (squares), P—postnatal day.

**Figure 3 cells-12-02839-f003:**
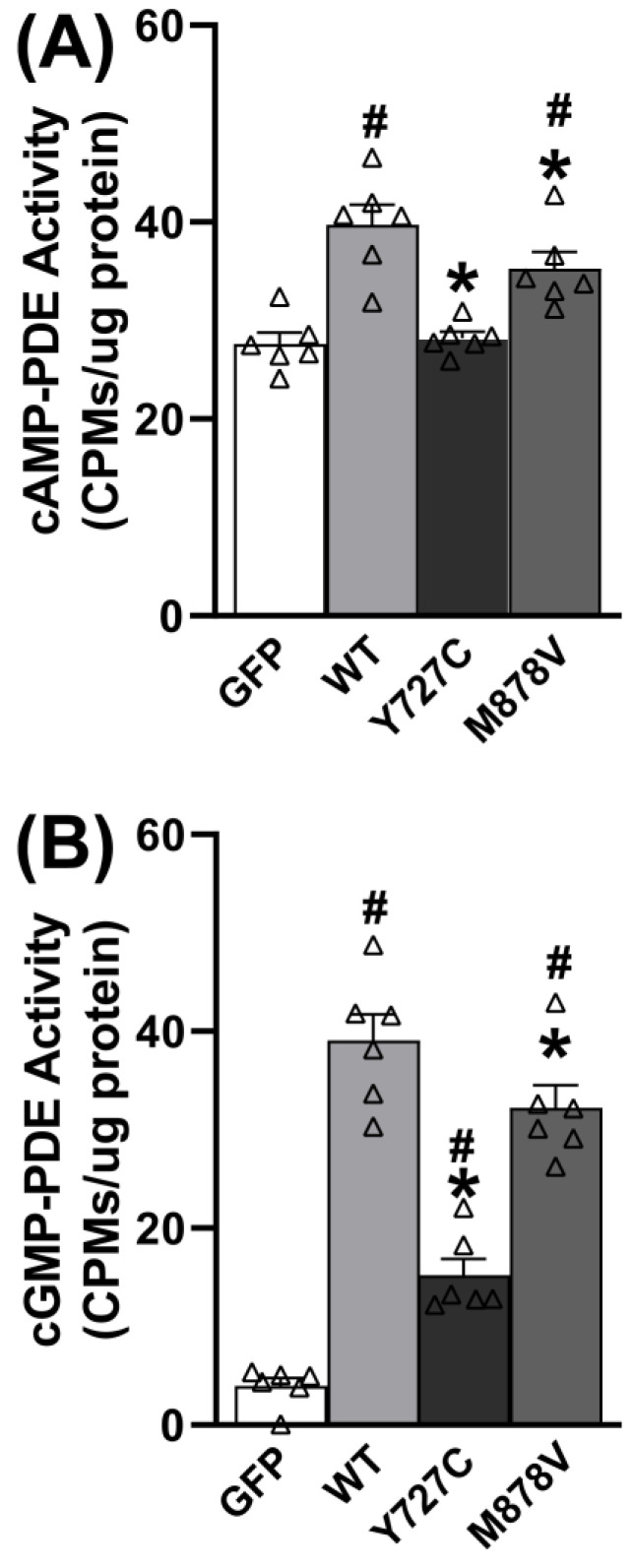
The Y727C and M878V PDE11A variants reduce cAMP and cGMP hydrolytic activity. (**A**) As expected, the hydrolysis of cAMP increases in HT22 cells when they are transfected with PDE11A4-WT relative to GFP. This PDE11A4-mediated hydrolysis of cAMP is completely eliminated by PDE11A4-Y727C but only reduced ~35% by PDE11A4-M878V. (**B**) Similarly, cGMP-PDE activity increases when HT22 cells are transfected with PDE11A4-WT relative to GFP, and PDE11A4-cGMP activity is reduced ~70% by PDE11A4-Y727C but only 20% by PDE11A4-M878V. Post hoc: * vs. WT, *p* = 0.0448–0.0002; # vs. GFP, *p* = 0.004–0.0001. ^Δ^ individual data point.

**Figure 4 cells-12-02839-f004:**
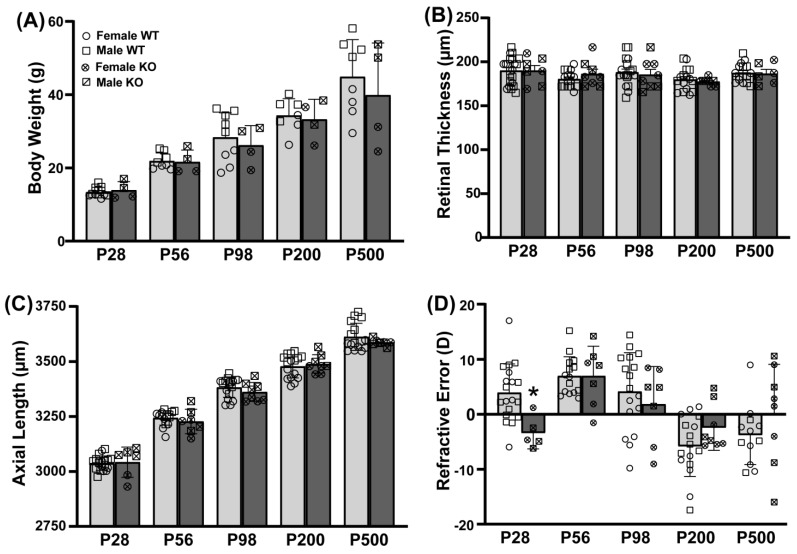
Eyes from Crispr B6/N *Pde11a* KO and WT male and female mice do not differ anatomically and only slightly differ on refractive error. Ocular biometry was performed on *Pde11a* KO eyes (n = 3–4 eyes from 2F + 3–4 eyes from 2 M per age) and *Pde11a* WT eyes (n = 8 eyes from 4 F at each age + 8 eyes from 4M at P56, P200, and P500, 13–14 eyes from 7 M at P28, and 9–10 eyes from 5 M at P98). (**A**) *Pde11a* KO mice show equivalent age-related increases in body weight relative to WT mice. (**B**) Neither retinal thickness nor (**C**) axial length differ between *Pde11a* KO and WT eyes. (**D**) Refractive error from *Pde11a* KO eyes (n = 2–4 eyes from 1-2F + 3–4 eyes from 2M per age) and *Pde11a* WT eyes (n= 7–8 eyes from 4 F at each age + 8 eyes from 4 M at P56 and P200, 12 eyes from 7 M at P28, 9 eyes from 5 M at P98, and 4 eyes from 2 M at P500) differs between genotypes at P28. Data expressed as mean ± SEM with individual data points plotted (◦ (circle), female; ▫ (square), male). * vs. WT *p* = 0.0122.

**Figure 5 cells-12-02839-f005:**
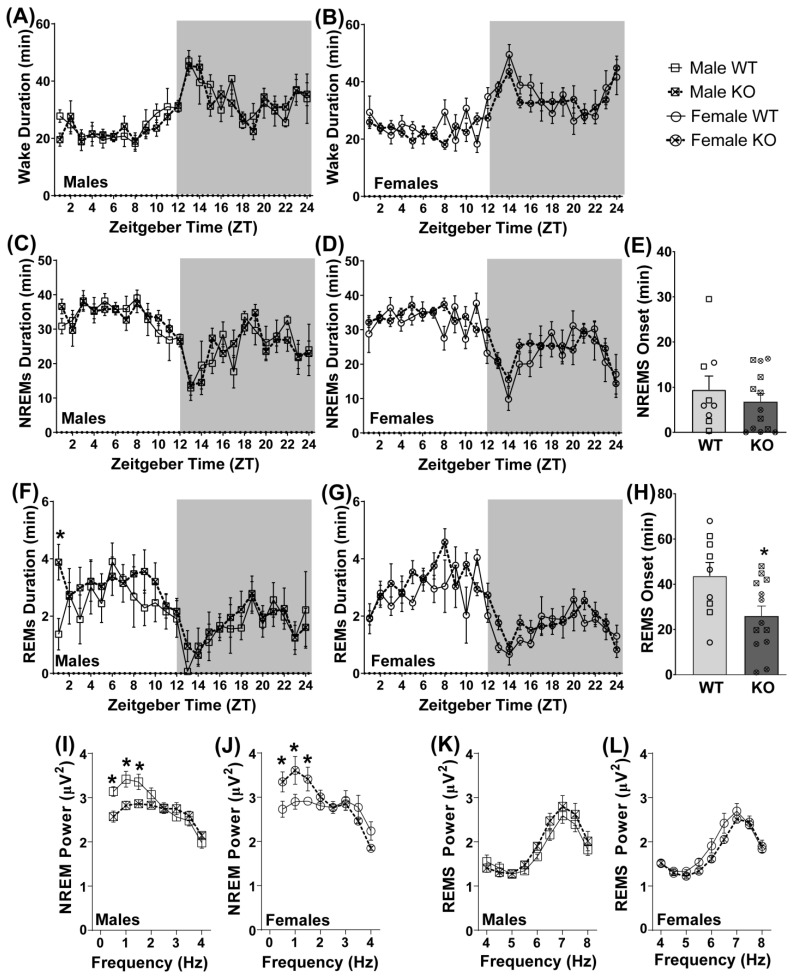
Differences between LacZ B6/J *Pde11a* KO and WT littermates on measures of a sleep–wake cycle are subtle and sex-specific. There is no significant difference between *Pde11a* KO and WT littermates on duration of waking in either (**A**) males (WT, n = 9; KO, n = 7) or (**B**) females (WT, n = 4; KO, n = 6; see Appendix A for bout#). There is also no significant difference between *Pde11a* KO and WT littermates on the duration of non-rapid eye movement sleep (NREMS) in (**C**) males or (**D**) females. (**E**) Onset of NREMS is also equivalent between genotypes. (**F**) Relative to WT male littermates, *Pde11a* KO male mice spend more time in rapid eye movement sleep (REMS) during the first hour of the light cycle. (**G**) No such genotype effect is observed on female REMS duration. (**H**) Across sexes, *Pde11a* KO mice exhibit a significantly shorter onset to REMS relative to WT littermates. Power at the lower frequencies within the delta range differ between genotypes as well, but with (**I**) male *Pde11a* KO mice showing reduced power relative to WT males and (**J**) female *Pde11a* KO mice showing increased power relative to WT females. A strong trend towards an opposing genotype effect in (**K**) male versus (**L**) female *Pde11a* KO mice is also observed across the higher frequencies of theta power during REMS. Post hoc: * vs. WT, *p* = 0.0263–0.0017. Data expressed as mean ± SEM with individual data points plotted over histograms (◦ (circle), female; ▫ (square), male).

**Figure 6 cells-12-02839-f006:**
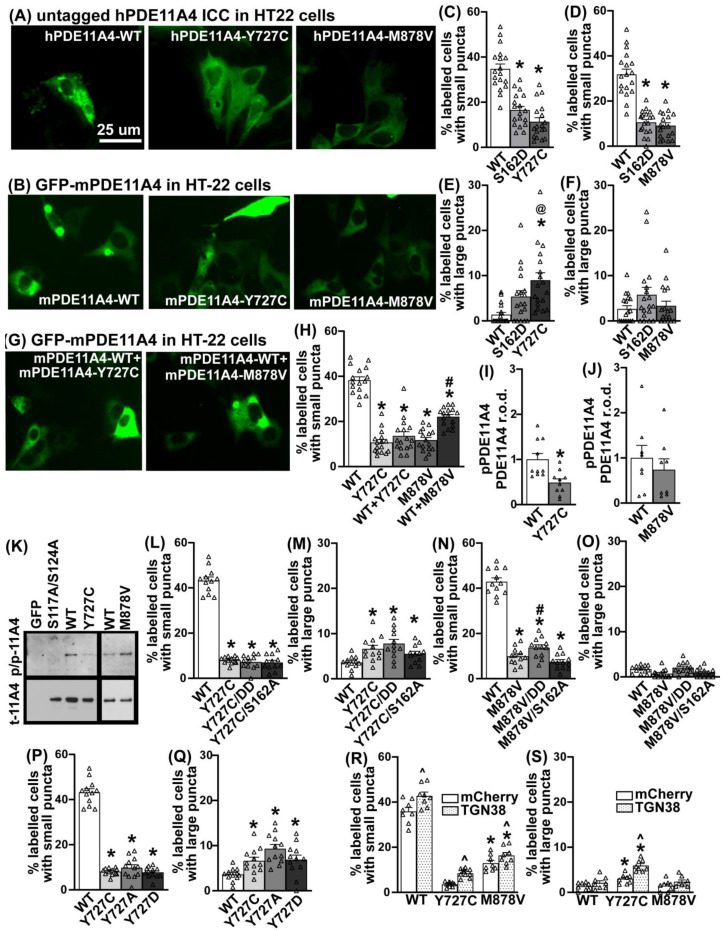
The Y727C and M878V variants dramatically alter PDE11A4 subcellular localization in HT22 cells. (**A**) Immunocytochemistry of untagged human PDE11A4 (hPDE11A4) or (**B**) direct visualization of a GFP-tagged mouse PDE11A4 (mPDE11A4) shows that the Y727C and M878V variants disperse the enzyme. Quantification shows that (**C**) Y727C and (**D**) M878V reduce the occurrence of small mPDE11A4 puncta to the same extent as a previously reported phosphomimic mutation of serine 162 (S162D [5]; each variant, n = 18 biological replicates/group over 3 experiments). (**E**) Y727C also increases the presence of larger mPDE11A4 punctate accumulations, while (**F**) S162D and M878V do not. (**G**,**H**) When the Y727C variant is co-expressed with PDE11A4-WT, as would occur in a patient, the phenotype of the Y727C variant completely dominates that of the WT construct, whereas WT+M878V results in the mathematically predicted 50% effect size of M878V alone. (**I**) Y727C decreases PDE11A4-pS117/pS124 (n = 5/group) but (**J**) M878V does not (n = 4/group). (**K**) Images of representative PDE11A4-pS117/pS124 (pp-11A4) and total PDE11A4 (t-11A4) Western blots. (**L**) Neither a phosphomimic S117D/S124D nor a phosphoresistant S162A changes the ability of Y727C to reduce the presence of small PDE11A4 puncta nor (**M**) increase the presence of large PDE11A4 accumulations. (**N**) Similarly, S162A does not alter the dispersal effects of M878V; however, M878V is not able to fully block the pro-accumulating effect of S117D/S124D. (**O**) There is no significant interaction between M878V and the phosphomutants on large PDE11A4 puncta. (**P**) The phosphoresistant Y727A and the phosphomimic Y727D mutants elicit the same effect as the Y727C variant on (**P**) small and (**Q**) large PDE11A4 puncta. (**R**) Relative to mCherry alone (i.e., negative control), overexpression of TGN38 increases the presence of small PDE11A4-WT, PDE11A4-Y727C, and PDE11A4-M878V puncta and (**S**) doubles the effect of Y727C on the accumulation of large PDE11A4 puncta. Histogram stretch, brightness, and contrast adjusted for graphical clarity. Post hoc * vs. WT, *p* = 0.0448–0.0002; # vs. GFP, *p* = 0.004–0.0001; @ vs. S162D, *p* < 0.05; ^ vs. mCherry, *p* = 0.001–0.0002. ^Δ^ individual data point.

**Table 1 cells-12-02839-t001:** Protein concentrations for each batch of tissue.

Batch	Segment	Sex	Acetone	Sample Concentration for Western Blot (µg/µL)
1	Anterior	male	yes	2.6
2	Anterior	female	no	2.32
3	Anterior	male	no	2.6
4	Anterior	female	no	2.44
1	Posterior	male	yes	3
2	Posterior	female	no	2.72
3	Posterior	male	no	3
4	Posterior	female	no	3
1	Retina	male	yes	2.44
2	Retina	female	no	2.44
3	Retina	male	no	2.6
4	Retina	female	no	2.44

## Data Availability

All data reported in this manuscript can be found in the Appendix A.

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
