# Peer review of "The Sleep Quality- and Myopia-Linked PDE11A-Y727C Variant Impacts Neural Physiology by Reducing Catalytic Activity and Altering Subcellular Compartmentalization of the Enzyme"

_cells, 2023, doi:10.3390/cells12242839_

Round 1

Reviewer 1 Report

Comments and Suggestions for Authors

Congratulations on your work, I only have some minor questions about it.

In the preparation of samples for Western blots, by "batch", do you mean that you pooled the samples?

In the assessment of cataracts, were the animals anesthetized?

Did you use a constitutive protein, like actin, in order to evaluate that the same protein amount was loaded in each gel?

On the other hand, I think you completed your objetives by the demonstration of the expression of PDE11A-Y7272C as well as the activity in order to further investigate therapeutic targets. 

This is a very complete work.

Author Response

Thank you kindly for the very supportive review. Please find a point-by-point address of your questions below:

1) With regard to preparation of Western blot samples, "batch" did not refer to pooling samples. "Batch" referred to a set of samples that included a balanced number of subjects/biological group that could all fit on one gel. We added this point of clarification to the Methods section.

2. Mice were not anesthetized for assessment of cataracts as experimenters were highly trained in mouse handling and sedation was not needed. We revised the Methods to clarify this point.

3. To confirm the same protein amount was loaded per sample, membranes were stained with Ponceau S as a loading control based on the best practice statement of the Journal of Biological Chemistry [50]. This is noted in the Methods section.

Reviewer 2 Report

Comments and Suggestions for Authors

The Y727C mutation in the phosphodiesterase PDE11A4 has been genetically linked to improved sleep quality and reduced risk for myopia. This prompted Sbornova et al. to analyse mice PDE11A4 knockout mutants and to further investigate PDE11A4(Y727C) with respect to enzymatic activity and its distribution within the cell.

First, the authors clearly showed that PDE11A4 is present in the eye, where it is particularly enriched in the retina. There, however, its amount decreases with increasing age until it obviously reaches a constant level at P200. Subsequently, they showed that, in HT22 cells, the Y727C mutation leads to a complete loss of cAMP phosphodiesterase activity and to a strong reduction of cGMP phosphodiesterase activity. Here, a detailed kinetic analysis of the purified wildtype and mutated enzyme including the determination of turnover number and Km constant would have been desirable and should definitely be done in the future.

While the loss of PDE11A4 does not seem to influence eye growth it seems to slightly improve sleep quality. Ultimately, the authors showed that the Y727C mutation strongly alters subcellular compartmentalisation by impacting processing via the trans-Golgi network. This effect is clearly due to the loss of tyrosine at this position.

The described experiments appear to be properly done. Although the presented results do not yet explain the phenotype caused by PDE11A4(Y727C), they are clear and constitute a basis for further analyses, which might help to establish PDE11A4 as a drug target.

Minor points and typos:

Page 4, Table 1: change “ug/ul” to “µg/µL”.

Page 5, 2.7. “PDE Assay”. The mode of operation of this assay does not become obvious at all from this section nor is it explained in the corresponding RESULTS section (page 12). At least a reference should be provided to give the reader a chance to understand how this assay works (e.g. Rybalkin et al. 2013. Methods Mol. Biol. 1020:51-62).

Page 12, lines 384 – 385: “PDE11A4-Y727 and … neural cells.” should be italic.

Page 13, line 406: change “… effect significant effect …” to “…significant effect …”.

Sometimes the authors write “COS1 cells” sometimes “COS-1” cells (e.g. page 19, lines 485 and 488). Please change to “COS-1 cells” throughout.

Page 21, lines 512 – 513: “The effects … S117/S124.” should be italic.

Page 22, lines 541 – 542: “The effects … network.” should be italic.

Page 24, line 678: change “PDE11AY-727C” to “PDE11A-Y727C”.

Author Response

Many thanks to the reviewer for the positive review and for taking the time to bring typos and other points for improvement to our attention! They were much appreciated! Below, please find a point-by-point discussion of how we changed the manuscript accordingly.

1) Reviewer 2 pointed out that a detailed kinetic analyses of WT vs. mutant PDE11A4 should be of interest to future studies. To address this comment, we added the following sentence to the Discussion: A detailed kinetic analysis of purified PDE11A4 versus PDE11A4-Y727C and PDE11A4-M878V will be of interest to future studies, including the determination of turnover number and Km constant.

2) Typo fixed in Table 1.

3) Reviewer 2 pointed out we neglected to describe the overall process of the PDE assay. To address this concern, the following sentence was added to the Methods section: PDE activity was measured using a coupled-enzyme assay that hydrolyzes [3H]cGMP or [3H]cAMP and then quantifies the resultant 5'-[3H]GMP and 5'-[3H]AMP that is isolated using low salt equilibrated anion-exchange columns (based on [52] with some modification as per [12]). 

4) Page 12 section heading is italicized.

5) "effect significant effect" changed to "significant effect"

6) Page 21 section heading is italicized

7) Page 22 section heading is italicized.

8) “PDE11AY-727C” changed to “PDE11A-Y727C”